# Assessment of mucin-related gene alterations following treatment with rebamipide ophthalmic suspension in Sjögren's syndrome-associated dry eyes

**Jun Shoji**[ID][☯], **Noriko Inada**[☯], **Akiko Tomioka**[☯], **Satoru Yamagami**[*][☯]

Division of Ophthalmology, Department of Visual Sciences, Nihon University School of Medicine, Tokyo, Japan

☯ These authors contributed equally to this work.
* satoru.yamagami@nihon-u.ac.jp

**Data Availability Statement:** All relevant data are within the paper and its Supporting Information files.

## Abstract

Ocular surface mucins are thought to play vital roles in maintaining the homeostasis of the pre-ocular surface tear film. We performed ocular surface tests with impression cytology to assess the expression levels of mucin-related genes on the ocular surface in healthy eyes. In addition, we investigated alterations in mucin-related gene expression secondary to treatment with rebamipide ophthalmic suspension in patients with Sjögren's syndrome-associated dry eyes (SS-DE). Thirty-three healthy individuals (control group) and 13 patients from our hospital with SS-DE were enrolled. Impression cytology was performed using Schirmer's test paper for RNA sampling. The mRNA levels of SAM-pointed domain-containing ETS-like factor (*SPDEF*), mucin 5AC (*MUC5AC*), and mucin 16 (*MUC16*) were determined using a real-time reverse transcription-polymerase chain reaction. The ocular surface test was performed once for the control group, and at baseline as well as 2, 4, 8, and 12 weeks after treatment in the Sjögren's syndrome-associated dry eyes group. mRNA levels of *SPDEF*, *MUC5AC*, and *MUC16* were not significantly different between the control and SS-DE groups before rebamipide ophthalmic suspension treatment. *SPDEF* mRNA levels in control subjects were significantly correlated with levels of *MUC5AC*. Among SS-DE patients, *SPDEF* mRNA levels were significantly increased at 2, 4, and 8 weeks after treatment compared with baseline levels. *MUC16* mRNA levels were significantly decreased from baseline levels at 4 and 8 weeks post-treatment. Ocular surface test using impression cytology is a clinically useful tool for assessing mucous conditions on the ocular surface and can be used to determine the effects of instillation treatment with eye drops that affect mucin production at the ocular surface.

## Introduction

Sjögren's syndrome is an autoimmune inflammatory disorder that leads to the development of dry mouth and eyes through functional impairment of the exocrine glands, specifically those

**Funding:** This study was supported by a grant from Otsuka Pharmaceutical Co., Ltd. to Nihon University Itabashi Hospital. Otsuka Pharmaceutical Co., Ltd. had no role in the study design, data collection and analysis, and the decision to publish this manuscript.

**Competing interests:** Financial support for this research was received from the Otsuka Pharmaceutical Co., Ltd., Japan. I have read the journal's policy and the authors of this manuscript have the following competing interests: J.S. received personal fees from Santen Pharmaceutical Co. Ltd., Senju Pharmaceutical Co. Ltd., Alcon Pharmaeuticals, outside the submitted work. Y.S., N.I., and A.T. declare that they have no conflict of interest. This does not alter our adherence to PLOS ONE policies on sharing data and materials.

that produce saliva and tears [1]. Patients with Sjögren's syndrome-associated dry eyes (SS-DE) exhibit symptoms such as dryness, discomfort, ocular irritation, and visual disturbance. SS-DE is accompanied by tear deficiency, increased tear film osmolarity [2,3], and ocular surface inflammation. The inflammation is caused by inflammatory cell infiltration, activation of the ocular surface epithelium with increased expression of cytokines and chemokines [4], and an increased concentration of inflammatory cytokines and chemokines in the tear fluid [5]. In recent studies, mucins have been implicated in the pathogenesis of dry eye disease [6,7].

Mucins are high molecular weight glycoproteins that contain a central region formed of tandem repeats and O-type carbohydrate chains bound to core proteins. In humans, 20 mucin genes (*MUC1*, *MUC2*, *MUC3A*, *MUC3B*, *MUC4*, *MUC5AC*, *MUC5B*, *MUC6* to *MUC13*, and *MUC15* to *MUC19*) are categorized based on their amino acid sequences [8]. The following mucins are found on the ocular surface and are involved in moisture retention and homeostatic maintenance: *MUC5AC* is a gel-forming mucin, *MUC1*, *MUC4*, and *MUC16* are membrane-associated mucins, and *MUC7* is a soluble mucin [9,10].

On the ocular surface, *MUC5AC* is secreted by goblet cells. The differentiation of goblet cells is controlled by SAM-pointed domain-containing ETS-like factor (*SPDEF*), a transcription factor also involved in the differentiation of goblet cells in the airway and intestinal epithelia. Furthermore, *SPDEF* is a key regulator of goblet cell hyperplasia and mucous hypersecretion in mucosal tissues [11–14]. A potential role for *SPDEF* in the pathophysiology of dry eye disease has been reported in a *SPDEF* knockout mouse model [15]. *MUC16*, a different mucin expressed in the ocular surface glycocalyx, has been shown to interact with the actin cytoskeleton in the microplicae of the ocular surface epithelium and to provide an anti-adhesive barrier [16,17]. Moreover, the morphological characteristics of the ocular surface in patients with dry eye disease are thought to include a reduction in goblet cells, squamous metaplasia, and keratinization of the ocular surface epithelium [18]. As a result of these morphological changes, the mucins expressed on the ocular surface are thought to be altered [6]. Therefore, assessments of mucin and mucin-related factors on the ocular surface are crucial clinical evaluations for determining the severity of dry eye disease.

Two novel eye drops that treat dry eye by promoting mucous secretion on the ocular surface, diquafosol and rebamipide, were launched in Japan in 2010 and 2012, respectively. These eye drops reportedly improve the subjective symptoms of dry eye [19,20]. Diquafosol is a $P2Y_2$ receptor agonist and promotes mucin secretion from conjunctival goblet cells, as well as fluid secretion from conjunctival epithelial cells that express $P2Y_2$ receptors at the cellular membrane [21,22]. Rebamipide is a novel synthesized quinolinone approved for the treatment of gastric ulcers and gastritis as an oral therapeutic drug, and to treat dry eye as an ophthalmic suspension. Rebamipide reportedly increases the number of goblet cells in the normal rabbit conjunctiva [23] and the lid wiper [24, 25] of human conjunctiva [26].

The primary objective of this study was to clarify the normal range of mucin-associated gene expression levels in the ocular surface of healthy volunteers using a cross-sectional analysis. To do so, we performed an ocular surface test using impression cytology with Schirmer's filter paper to assess mucin-related gene expression on the ocular surface in healthy subjects and patients with SS-DE. The normal range was by the calculated mean values and 95% confidence intervals for each mucin-related gene expression level.

Our secondary objective was to investigate potential treatment-induced alterations of mucin-related gene expression levels in SS-DE patients. To do so, we evaluated alterations in *SPDEF*, *MUC5AC*, and *MUC16* gene expression levels in SS-DE patients at three follow-up points after treatment with rebamipide. The alteration of mucin-related gene expression levels at each observational point was evaluated in comparison with baseline values.

## Materials and methods

### Subjects

This prospective, cross-sectional study was approved by the institutional review board of the Nihon University Itabashi Hospital, Tokyo, Japan (RK-150310-10, UMIN: 000017620) and adhered to the tenets of the Declaration of Helsinki. Written informed consent was obtained from all participants. The control group consisted of 33 healthy adult volunteers (7 men and 26 women) without ocular disorders and who did not wear contact lenses. Thirteen patients being treated at the hospital were included in the SS-DE group. The inclusion criterion was a diagnosis with Sjögren's syndrome according to the revised Japanese criteria (1999); these included having a baseline Schirmer I test score < 5 mm as well as staining positive for fluorescein [27]. The exclusion criteria for SS-DE patients were as follows: (1) ocular surface disease (infectious blepharitis, lagophthalmos, blepharospasm, iritis, and conjunctival chalasis) other than dry eye; (2) treatment with topical anti-inflammatory drugs, such as corticosteroids or immunosuppressants; (3) the presence of punctal plugs or a history of surgical punctal occlusion; (4) a history of ocular surface surgery during the 12 months prior to study initiation, or intraocular surgery during the 3 months prior to study initiation; (5) use of contact lenses; and (6) previous systemic treatment for Sjögren's syndrome.

We prescribed a 4× daily dose of rebamipide ophthalmic suspension (Mucosta® ophthalmic suspension unit dose 2%; Otsuka Pharmaceutical Co., Ltd., Tokyo, Japan) to SS-DE patients. Clinical outcomes and mucin-related gene expression on the ocular surface of these patients were monitored for 12 weeks.

### Clinical examination of patients with dry eye

Both eyes of SS-DE patients were examined. All SS-DE patients underwent the tear breakup time test (TBUT) and fluorescein staining score evaluation, at baseline as well as 2, 4, 8, and 12 weeks after the initiation of rebamipide instillation therapy.

For fluorescein staining, one drop of saline solution was dropped onto the tip of the fluorescein test paper (FLUORES® Ocular Examination Test paper 0.7 mg, AYUMI Pharmaceutical Corporation, Tokyo, Japan), and then after application of fluorescein dye using fluorescein paper into the lower conjunctival sac, TBUT and fluorescein staining scores were evaluated under cobalt blue light using a slit lamp biomicroscopy (Haag-Streit Original Slit Lamp 900®BQ, Haag-Streit AG, Bern, Switzerland).

TBUT was measured as the time in seconds until the first appearance of a dark area in the fluorescein-stained precorneal tear film when the eye was kept open. The measurement was taken three times using a stopwatch, and the average time was calculated. The normal range for TBUT was considered to be 5 seconds or longer.

For the calculation of fluorescein staining score, the van Bijesterveld scoring system [28] was used. To assess epithelial damage, the temporal bulbar conjunctiva, cornea, and nasal bulbar conjunctiva were evaluated with a score of 0 (none), 1 (mild), 2 (moderate), and 3 (severe) in each area. The sum of the scores from the three areas was designated as the fluorescein staining score (Fig 1A).

### *SPDEF*, *MUC5AC*, and *MUC16* gene expression levels on the ocular surface

**Sample collection.** Impression cytology with filter paper for Schirmer's test was used to obtain RNA samples for real-time reverse transcription-polymerase chain reaction (RT-PCR) analysis [29,30]. The right eye of each participant in the control group and both eyes of SS-DE

a

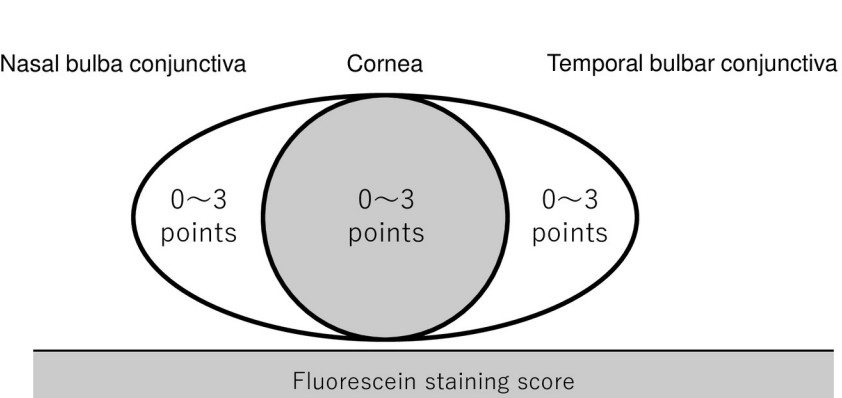

| Fluorescein staining score | | | | |
|---|---|---|---|---|
| Score | 0 | 1 | 2 | 3 |
| ① Nasal bulbar conjunctiva | None | Mild | Moderate | Severe |
| ② Cornea | None | Mild | Moderate | Severe |
| ③ Temporal bulbar conjunctiva | None | Mild | Moderate | Severe |
| Fluorescein staining score = ① + ② + ③ | | | | |

b

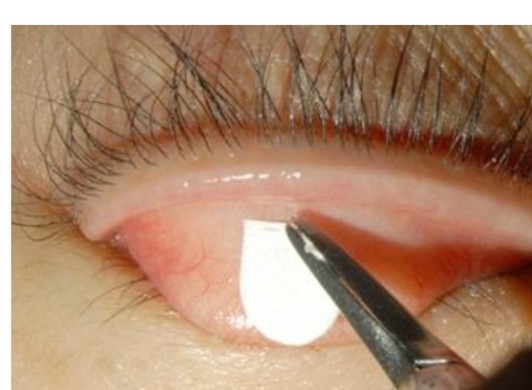

**Fig 1. Dry eye clinical examination.** (a) Criteria for determination of the fluorescein staining score. All sub-scores for the nasal and temporal bulbar conjunctiva and the cornea were summed, and the value designated as the total score. (b) Impression cytology was performed to obtain the sample from the upper palpebral conjunctiva.

patients were analyzed. Impression cytology was performed at baseline in the control group, and at baseline as well as 2, 4, 8, and 12 weeks in the SS-DE group.

Impression cytology was performed using the following technique: the 5 mm tip of Schirmer's test paper (Tear Production Measuring Strips, AYUMI Pharmaceutical Corporation, Tokyo, Japan) was placed at the center of the upper palpebral conjunctiva (Fig 1B) for 5 seconds using Beaupre cilia forceps (Inami & Co., Ltd, Tokyo, Japan). The test paper was then removed and preserved in microtubes containing RNAlater RNA Stabilization Reagent (Qiagen, Hilden, Germany) at 4°C until mRNA extraction.

**Real-time RT-PCR.** Total RNA from each Schirmer's test paper was extracted using the RNeasy® Mini Kit (Qiagen) following the manufacturer's instructions. cDNA was then synthesized using a High-Capacity cDNA Reverse Transcription Kit (Thermo Fisher, Waltham, Massachusetts, USA) according to the manufacturer's instructions.

To detect the expression of *SPDEF*, *MUC5AC*, and *MUC16* mRNA, real-time RT-PCR was performed using a commercial PCR master mix (TaqMan Universal PCR Master Mix; Thermo Fisher) and predesigned primers: *SPDEF* (Hs01026050_ml), *MUC5AC* (Hs00873651_m1), and *MUC16* (Hs01065189_m1; Thermo Fisher). Samples were analyzed using the Step One Plus real-time PCR system (Thermo Fisher), and comparative threshold (Ct) values were obtained. Target Ct values were normalized to those of glyceraldehyde 3-phosphate dehydrogenase (GAPDH) (Hs99999905_m1) from the same sample. The gene expression data were analyzed using the ΔΔCt method.

## Statistical analysis

The mRNA expression levels of *SPDEF*, *MUC5AC*, and *MUC16* are presented as mean with a 95% confidence interval (95% CI). The baseline associations in control group between participant's age and mucin-related gene expression, including *SPDEF*, *MUC5AC*, and *MUC16* mRNA levels, were assessed by partial correlation coefficient. The expression levels of mucin-related genes between men and women in the control group were compared using the Mann-Whitney U-test. Thirteen control subjects were age- and sex-matched to SS-DE patients, with random selection of controls if more than one matched with an SS-DE patient. Results of gene expression levels at baseline were compared between the matched SS-DE and control patients using the Mann-Whitney U-test. In the SS-DE group, comparisons between ration changes from baseline at baseline and 2, 4, 8, and 12 weeks after rebamipide treatment were analyzed using mixed-effects models. Because the patients with SS-DE may have a bilateral difference in disease severity, both eyes were analyzed to prevent a possible selection bias. A p-value < 0.05 was considered statistically significant. IBM SPSS Advanced Statistics software version 22 (IBM Corp.; Armonk, NY, USA) was used for statistical analysis.

# Results

## Expression levels of *SPDEF*, *MUC5AC*, and *MUC16* mRNA in the control group

The demographic characteristics of the 33 participants in the control group are shown in Table 1.

In the control group, we investigated the relationship between the age of the participants and expression levels of mucin-related genes on the ocular surface using the partial correlation method. There were no mucin-related genes that significantly correlated with the age of the participants (Table 2). Among the mucin-related genes, a significant correlation was found between *MUC5AC* and *SPDEF* mRNA expression levels (Fig 2 and Table 2). Additionally, we compared the mucin-related gene expression levels between male and female participants. *MUC5AC* and *MUC16* mRNA expression levels in men were significantly higher than in women (Table 3). However, their *SPDEF* mRNA expression levels were not statistically different.

## Alteration of clinical test results and *SPDEF*, *MUC5AC*, and *MUC16* mRNA expression levels on the ocular surface by rebamipide instillation in the SS-DE group

The demographic and clinical characteristics of the SS-DE group are shown in Table 4. The SS-DE group included 13 patients, all women; the mean age was 59.5 ± 10.6 years

**Table 1. Demographic and clinical characteristics of participants in the control group.**

| | | Control group |
|---|---|---|
| Subjects (eyes) | | 33 |
| Age (years) (mean ± SD) | | 61.8 ± 17.0 |
| Sex (men: women) | | 7: 26 |
| Mucin-related gene expression (relative expression levels) | *SPDEF* [mean (95% CI)] | 4.2 (2.8–5.6) |
| | *MUC5AC* [mean (95% CI)] | 1.9 (0.9–3.0) |
| | *MUC16* [mean (95% CI)] | 0.4 (0.3–0.6) |

CI, confidence interval; *SPDEF*, SAM-pointed domain-containing ETS-like factor.

**Table 2. Partial correlation coefficient between age and mucin-related gene expression levels in the control group.**

|         | *MUC5AC* | *MUC16* | *SPDEF* | Age |
|---------|----------|---------|---------|-----|
| *MUC5AC* | 1 | | | |
| *MUC16* | 0.058 | 1 | | |
| *SPDEF* | 0.733** | 0.328 | 1 | |
| Age | -0.089 | 0.098 | -0.067 | 1 |

** p = 2.7E-06.

(mean ± SD). Sjögren's syndrome classification levels (primary versus secondary), and the systemic anti-inflammatory treatments for the primary disease of patients with secondary SS-DE, are also shown in Table 4. The rebamipide eye drops were supplemented with sodium hyaluronate in three patients, and with artificial tears and sodium hyaluronate in one; no additional medications were added for nine patients. The compositions of the eye drops were not changed during the observation period. In the SS-DE group, expression levels of *SPDEF*, *MUC5AC*, and *MUC16* mRNA were 5.0 (1.9–8.2) [mean (95% CI)], 3.0 (1.2–4.9), and 0.7 (0.5–1.0), respectively. There were no statistical differences between the baseline levels for SS-DE patients and the age- and sex-matched controls (Table 4).

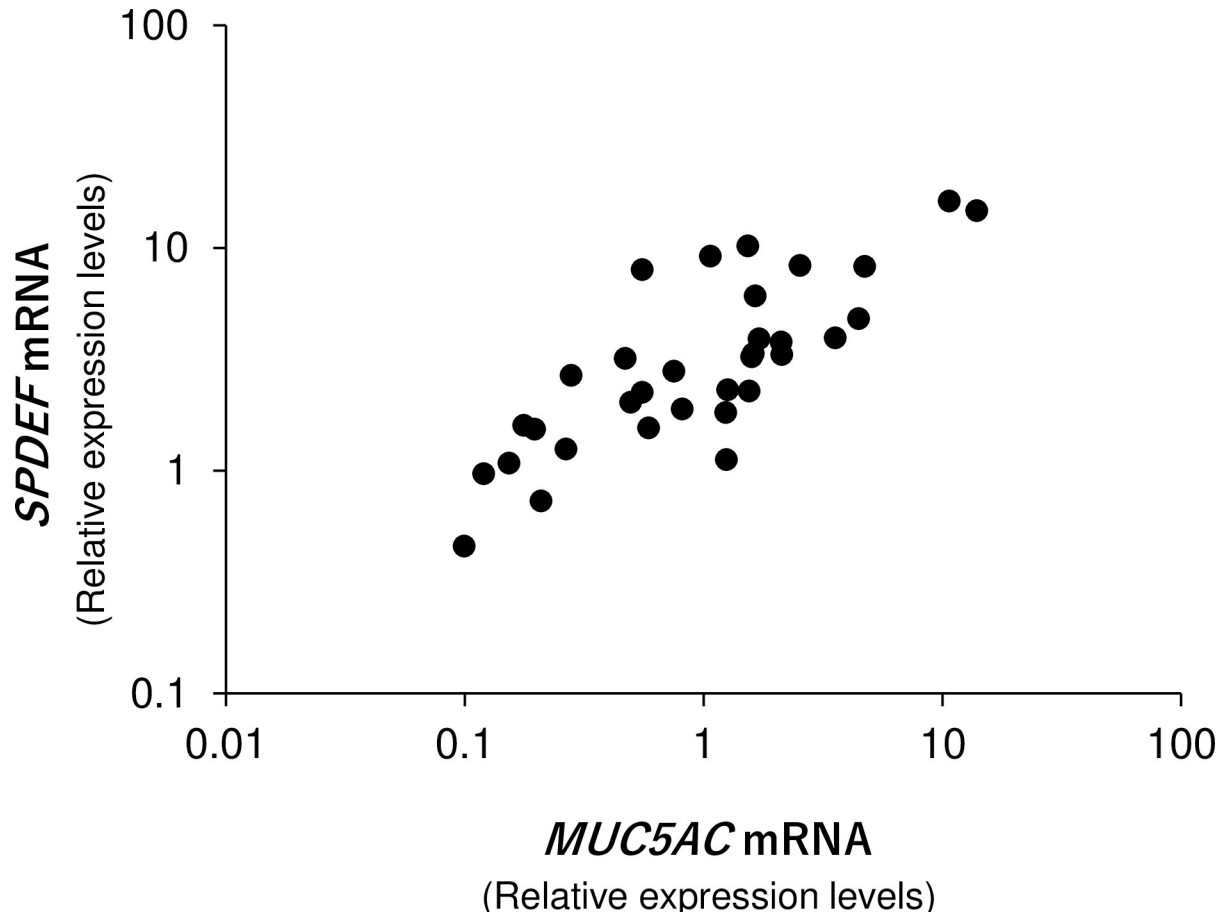

**Fig 2. Correlation between *SPDEF* and *MUC5AC* mRNA expression levels in the control group.** *SPDEF* mRNA expression levels were significantly correlated with those of *MUC5AC*. ρ = 0.791, p < 0.001, Spearman's correlation coefficient.

**Table 3. Comparison of mucin-related gene expression levels between male and female controls.**

| | Male controls | Female controls | *p*-value |
|---|---|---|---|
| Age [years (mean ± SD)] | 54.7 ± 19.4 | 63.7 ± 16.1 | 0.243 |
| *MUC5AC* mRNA [relative expression levels (mean ± SD)] | 4.7 ± 5.3 | 1.2 ± 1.2 | **0.038** |
| *MUC16* mRNA [relative expression levels (mean ± SD)] | 0.76 ± 0.34 | 0.35 ± 0.22 | **0.004** |
| *SPDEF* mRNA expression levels [relative expression levels (mean ± SD)] | 7.4 ± 6.3 | 3.4 ± 2.6 | 0.094 |

Regarding clinical examination, the fluorescein staining scores at 4 and 8 weeks after the initiation of rebamipide treatment were significantly decreased compared with baseline (p = 0.0005 and p < 0.0001 at 4 and 8 weeks, respectively; mixed-effects models; Fig 3A). In contrast, TBUT results at 2, 4, and 8 weeks after the initiation of rebamipide treatment were significantly increased compared with baseline (p = 0.0005, 0.0383, and 0.0020 at 2, 4, and 8 weeks, respectively; mixed-effects models; Fig 3B).

When *SPDEF*, *MUC5AC*, and *MUC16* mRNA expression levels on the ocular surface of patients were examined, we found that *SPDEF* mRNA expression levels at 2, 4, and 8 weeks after the initiation of rebamipide treatment had increased significantly (p = 0.0350, 0.0012, and 0.0069 at 2, 4, and 8 weeks, respectively, mixed-effects models; Fig 4A), but not at 12 weeks. Additionally, *MUC5AC* mRNA expression levels were significantly increased at the 8-week time-point (p = 0.0330, mixed-effects model; (Fig 4B). In contrast, *MUC16* mRNA expression levels at both 4 and 8 weeks after the initiation of rebamipide treatment had decreased significantly (p = 0.0358 and 0.0050 at 4 and 8 weeks, respectively, mixed-effects models; Fig 4C).

In accordance with an evaluation of ration changes from baseline in *SPDEF*, *MUC5AC*, and *MUC16* mRNA expression levels by mixed-effect model analysis, the 4-, and 8-week time-point was the observation periods with significant alterations in *SPDEF*, *MUC5AC*, and *MUC16* mRNA expression levels in SS-DE patients under rebamipide treatment. We

**Table 4. Demographic and clinical characteristics of patients with Sjögren's syndrome.**

| | | SS-DE patients | Age- and sex-matched controls | *p*-value |
|---|---|---|---|---|
| Subjects (cases) | | 13 | 13 | |
| Age [years, (mean ± SD)] | | 59.5 ± 10.6 | 57.5 ± 16.0 | 0.166 |
| Sex (male: female) | | 0: 13 | 0: 13 | |
| Immunological disease | | Primary SS: 3 cases | | |
| | | Secondary SS: 10 cases (RA: 5 cases, SLE: 2 cases, others: 3 cases) | | |
| Systemic anti-inflammatory treatment for primary disease patients with secondary SS | | Prednisolone: 3 cases | | |
| | | Mizoribine: 2 cases | | |
| | | Salazosulfapyridine: 1 case | | |
| | | Non: 4 cases | | |
| Baseline mucin-related gene expression (relative expression levels) | *SPDEF* [mean (95% CI)] | 5.0 (1.9–8.2) | 3.6 (2.1–5.1) | 0.489 |
| | *MUC5AC* [mean (95% CI)] | 3.0 (1.2–4.9) | 1.1 (0.6–1.6) | 0.522 |
| | *MUC16* [mean (95% CI)] | 0.7 (0.5–1.0) | 0.4 (0.2–0.5) | 0.106 |

RA, rheumatoid arthritis; SLE, systemic lupus erythematosus; SS, Sjögren's syndrome; SS-DE, Sjögren's syndrome-associated dry eyes.

CI, confidence interval; *SPDEF*, SAM-pointed domain-containing ETS-like factor.

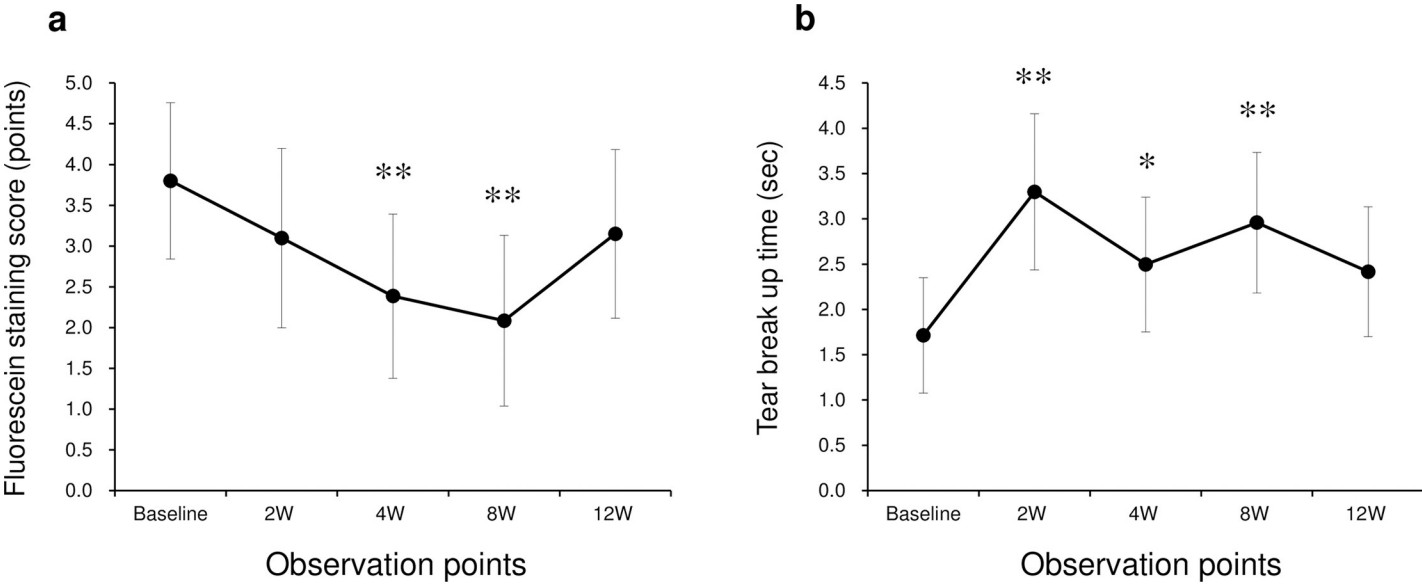

**Fig 3. Change in fluorescein staining scores and tear breakup time (TBUT) at follow-up after rebamipide instillation treatment in SS-DE patients.** Least-square means (LS means ± 95% confidence interval) ware plotted for fluorescein staining scores and tear breakup times (TBUT) at each observation point. (a) Fluorescein staining scores at 4 and 8 weeks were significantly decreased compared with baseline values. (b) TBUT at 2 and 8 weeks were significantly increased compared with baseline levels. *, $p < 0.05$; **, $p < 0.01$, mixed-effects models.

compared relative expression levels of *SPDEF*, *MUC5AC*, and *MUC16* mRNA in SS-DE group with those of the age- and sex-matched control group. At week 4, *MUC5AC* mRNA expression levels in the SS-DE group were significantly increased compared with those in age- and sex-matched controls (Fig 5A). In contrast, there were no significant differences in *SPDEF* and *MUC16* mRNA expression between the SS-DE and age- and sex-matched control groups (Fig 5B and 5C). At week 8, there was no significant difference between the SS-DE and control groups for all mRNA expression levels.

A representative patient is shown in Fig 6—a 77-year-old woman with SS-DE suffering from severe dryness and foreign body sensation in both eyes. She received instilled sodium hyaluronate ophthalmic solution 0.3% and rebamipide ophthalmic suspension 2%, 4 times daily in each eye. At the baseline examination, the fluorescein score was 4 points for both eyes. Photographs of her right eye with fluorescein staining are shown in Fig 4A–4D. TBUT of the right and left eyes at baseline were both 2 sec. The measured expression levels of *SPDEF*, *MUC5AC*, and *MUC16* mRNA in the right eye are shown in Fig 4E. Over 8 weeks of follow-up, her superficial punctate keratitis improved and there were increased expression levels of *SPDEF*, *MUC5AC*, and *MUC16* mRNA.

## Discussion

In this study, we investigated the utility of the ocular surface test, which clinically evaluates mucin-related gene (*SPDEF*, *MUC5AC*, and *MUC16*) expression levels on the ocular surface in patients with dry eye. Impression cytology is thought to be a suitable tool for obtaining genetic information on ocular surface biomarkers [31,32]. Our study found the following: 1) expression levels of *SPDEF* mRNA were significantly correlated with those of *MUC5AC* mRNA in healthy controls; 2) expression levels of *SPDEF* mRNA were significantly increased at 2, 4, and 8 weeks after rebamipide ophthalmic suspension treatment in SS-DE patients; and 3) expression levels of *MUC16* mRNA were significantly decreased at 4 and 8 weeks after rebamipide

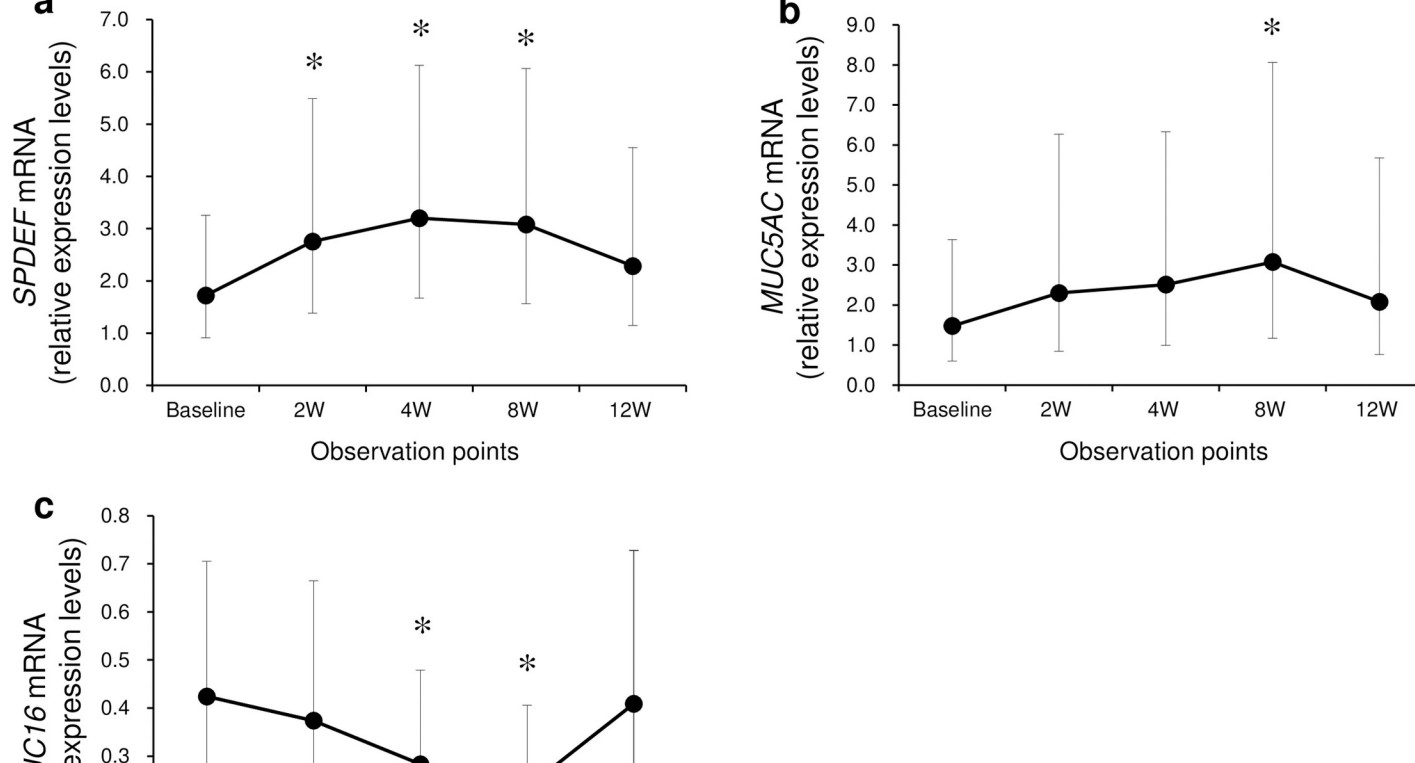

**Fig 4. Change of *SPDEF*, *MUC5AC*, and *MUC16* mRNA expression levels at follow-up points after rebamipide instillation treatment in SS-DE patients.** Least-square means (LS means ± 95% confidence interval) ware plotted for relative expression levels of *SPDEF*, *MUC5AC*, and *MUC16* mRNA at each observation point. (a) *SPDEF* mRNA expression levels were significantly increased at 2, 4, and 8 weeks compared with baseline values. (b) *MUC5AC* mRNA expression levels were significantly increased at 8 weeks compared with those at baseline. (c) *MUC16* mRNA expression levels were significantly decreased at 4 and 8 weeks compared with those at baseline. * p < 0.05, mixed-effects models.

ophthalmic suspension treatment. In addition, we found significant improvement over time in the fluorescein staining scores (4 and 8 weeks post-treatment) and in TBUT (2, 4, and 8 weeks post-treatment).

In the control group, mucin-related gene expression levels were not correlated with participants' age, implying that age does not affect the results of ocular surface tests for mucin. In contrast, *MUC5AC* and *MUC16* mRNA levels were significantly higher in men than in women. In a previous report, Corrales et al. did not find sex-specific differences in *MUC1*, *MUC2*, *MUC4*, *MUC5AC*, and *MUC7* gene expression levels in normal human conjunctiva [33]. However, their finding of significantly higher conjunctival goblet cell density from conjunctival impression cytology samples in men than in women [34,35] is consistent with our results of *MUC5AC* gene expression.

We observed a significant correlation between *SPDEF* and *MUC5AC* mRNA levels in our control group. *SPDEF* is thought to be a transcriptional factor involved in the differentiation of goblet cells and the glycosylation of mucin [36]. Furthermore, goblet cell count and goblet

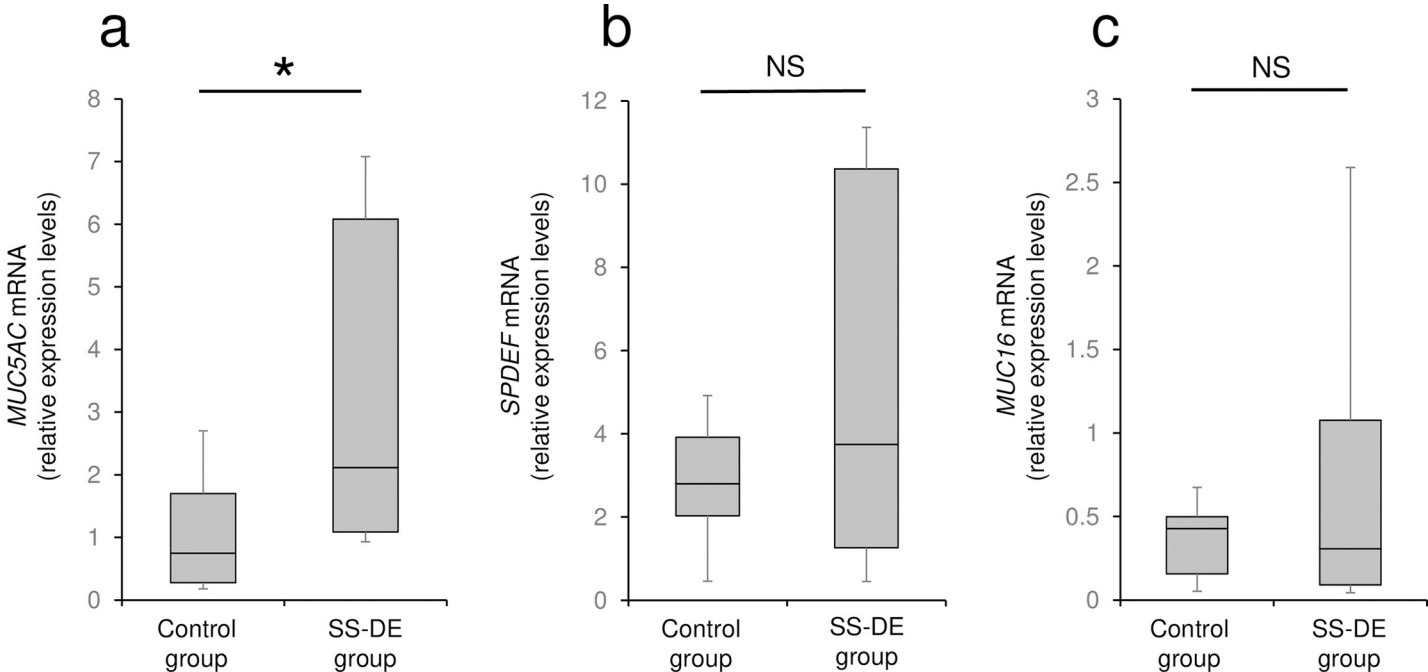

**Fig 5. Mucin-related gene expression levels in the SS-DE and age- and sex-matched control groups at 4 weeks after the initiation of rebamipide instillation treatment.** Mucin-related gene expression levels in the SS-DE at 4 weeks after the initiation of rebamipide treatment were compared with those in age- and sex-matched control groups. *MUC5AC* mRNA expression levels in the SS-DE group were significantly higher than in the control group. (a) *MUC5AC* mRNA expression; (b) *SPDEF* mRNA expression; (c) *MUC16* mRNA expression. * p < 0.05, Mann-Whitney U-test. SS-DE, Sjögren's syndrome-associated dry eyes.

cell-associated genes (e.g., *MUC5AC*, *GCNT3*, and *TFF1*) are reportedly reduced in the conjunctiva of *SPDEF*$^{-/-}$ mice [15]. *MUC5AC* on the ocular surface is thought to be of goblet cell origin [37]. Therefore, *SPDEF* mRNA expression levels on the ocular surface may reflect conjunctival goblet cell count.

Baseline *SPDEF*, *MUC5AC*, and *MUC16* mRNA expression levels in SS-DE patients were not significantly different from those in age- and sex-matched controls. Interestingly, previous studies have reported that patients with Sjögren's syndrome have decreased *MUC5AC* in tears and in *MUC5AC* mRNA expression compared with control subjects [32,38]. These opposing results may be due to several reasons. First, the reported decrease in *MUC5AC* mRNA levels was in the context of impression cytology samples obtained from the bulbal conjunctivae of SS-DE patients [31], while our impression cytology samples were obtained from the upper palpebral conjunctivae. In clinical practice, rose bengal and fluorescence staining in patients with dry eye disease is generally more intense in the bulbar conjunctivae than in the palpebral conjunctivae. Therefore, our samples obtained from palpebral conjunctivae may not completely capture the expected alterations in mucin-related gene expression in SS-DE patients. Second, the basic therapeutic drugs typically used for primary disease in SS-DE patients may be the reason for their normal baseline expression levels of *MUC5AC* mRNA. The use of steroids and other immunosuppressants by our SS-DE patients may have led to improvement in their *MUC5AC* expression profiles, since cyclosporine eye drops are also reportedly effective for the treatment of dry eye disease [39,40].

In the clinical examinations, we found that both the fluorescein staining scores and TBUT improved in the SS-DE group between 4 and 8 weeks after the initiation of treatment. These results indicate that rebamipide instillation improves clinical outcomes in patients with dry eye, as in previous reports [41].

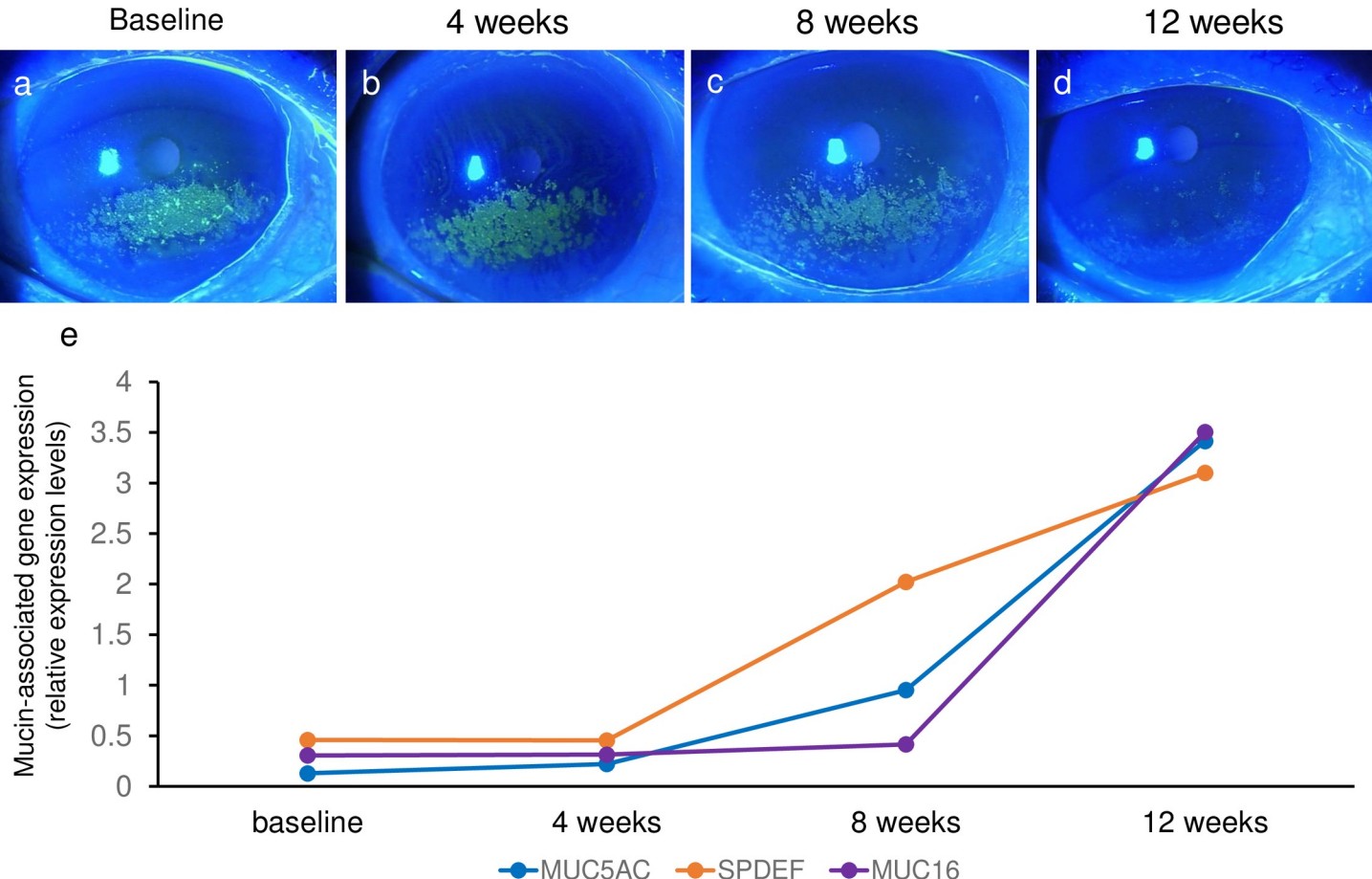

**Fig 6. A 77-year-old women with SS-DE disease.** Photographs of cornea with fluorescein staining at baseline (a), and 4 (b), 8 (c), and 12 (d) weeks. (e), expression levels of *SPDEF*, *MUC5AC*, and *MUC16* mRNA in the right eye measured by reversed transcription-polymerase chain reaction.

We assessed the effects of rebamipide ophthalmic suspension on mucin-related gene expression on the ocular surface of patients with SS-DE. During the observational period from weeks 2–8 after the start of rebamipide therapy, *SPDEF* and *MUC5AC* mRNA expression levels increased while *MUC16* levels decreased. Previous studies have demonstrated that rebamipide increases the number of goblet cells in normal rabbit conjunctiva [23] as well as in the lid wiper of human conjunctiva [26]. Therefore, the *SPDEF* expression level increase that we observed in the ocular surface test may also reflect an increase in conjunctival goblet cells. Furthermore, the significant increase in *MUC5AC* mRNA expression levels at 8 weeks after baseline may also be due to an increase in goblet cells in the conjunctiva.

The effect of rebamipide ophthalmic suspension on ocular surface *MUC16* expression is controversial. In previous studies, rebamipide increased *MUC16* gene expression in cultured corneal epithelial cells and in a murine model of primary Sjögren's syndrome [42,43]. Furthermore, we have previously reported two dry eye cases for which *MUC5AC* and *MUC16* mRNA expression on the ocular surface was increased by the concomitant use of rebamipide and steroid ophthalmic suspensions, although the increase in *MUC16* mRNA levels was lower than that of *MUC5AC* mRNA levels [44]. In contrast, Takeji et al. reported that rebamipide increased *MUC1* and *MUC4* but not *MUC16* gene expression in cultured human corneal

epithelial cells [45]. Gipson et al. have reported that *MUC5AC* and MUC16 are expressed in goblet cells and that these cells might be a second source of *MUC16* in the conjunctiva tissue [46]. *MUC16* has been associated with *SPDEF* expression in an *SPDEF* null mouse [15, 47] and correlated with *MUC5AC* in goblet cells within human conjunctiva [46]. In the current study, *MUC16* mRNA expression levels at 4 and 8 weeks after the initiation of rebamipide treatment were significantly decreased compared with those at baseline. However, at 8 weeks of follow-up, there was no statistically significant difference in *MUC16* mRNA expression levels between SS-DE patients and controls. In future studies of ocular surface testing, simultaneous measurement of the expression levels of *MUC5AC, MUC16, and SPDEF* mRNA, as well as inflammation-related factors, such as cytokines, is needed to gain a better understanding of *MUC16* expression on the ocular surface.

Clinical examinations and ocular surface tests conducted at week 12 after treatment found that all parameters were trending back toward baseline levels. The reasons for this result is unknown. It is possible that some patients in the SS-DE group did not adhere to the instillation therapy regimen, either because of improvement in their symptoms or because they forgot to apply the treatment. The possibility that uneven continuation may be due to difficulties administering the drops at work. Thus, once an adequate treatment dosage is determined, ongoing monitoring of mucin expression levels in the ocular surface is needed to assess the long-term effectiveness of the treatment.

Our study had two main limitations. First, this study lacked a control group treated with a placebo containing only the vehicle used to prepare the ophthalmic suspension. The potential influence of different eye drop vehicles on ocular surface mucin expression will be investigated in a future study. The effect of the rebamipide ophthalmic suspension on ocular surface mucin and the barrier function of conjunctival and corneal epithelium should also be determined in healthy individuals and in patients with non-dry eye conjunctivitis. Second, a long-term follow-up period, controlling for strict adherence, was not included. To more precisely determine mucous conditions and the effects of rebamipide treatment, future mucin-related gene expression studies should be conducted with a substantially greater number of subjects and stricter monitoring of adherence.

## Conclusions

The mucin-related gene expression assay presented in this study may be a useful tool to evaluate tear film formation on the ocular surface of patients with SS-DE. The ocular surface test has the potential to become a leading test for dry eye, and may eventually allow mucin-promoting eye drops to become a standard medical treatment.

## Supporting information

**S1 Table. The raw data of control individuals.**
(XLSX)

**S2 Table. The raw data of the patients with Sjögren syndrome-associated dry eye.**
(XLSX)

## Acknowledgments

We would like to thank Mr. Toshihito Furukawa of Biostatistical Research Co. Ltd, Tokyo, Japan, for his contribution to the statistical analyses.

## Author Contributions

**Investigation:** Jun Shoji, Noriko Inada, Akiko Tomioka.

**Methodology:** Jun Shoji.

**Writing – original draft:** Jun Shoji.

**Writing – review & editing:** Satoru Yamagami.

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
