## [Decision Letter · Decision Letter 0]

14 Sep 2020

PONE-D-20-24941

A method for the assessment of mucin-related gene alterations following treatment with rebamipide ophthalmic suspension in Sjögren’s syndrome-associated dry eyes

PLOS ONE

Dear Dr. Yamagami,

Thank you for submitting your manuscript to PLOS ONE. After careful consideration, we feel that it has merit but does not fully meet PLOS ONE’s publication criteria as it currently stands. Therefore, we invite you to submit a revised version of the manuscript that addresses the points raised during the review process.

Both expert reviewers requested more explanation of your methods, and both asked for a better explanation of how controls were sex matched with Sjögren Syndrome subjects.

We look forward to receiving your revised manuscript.

Kind regards,

Alfred S Lewin, Ph.D.

Academic Editor

PLOS ONE

Journal Requirements:

2. PLOS ONE requires experimental methods to be described in enough detail to allow suitably skilled investigators to fully replicate and evaluate your study. See https://journals.plos.org/plosone/s/submission-guidelines#loc-materials-and-methods for more information.

To comply with PLOS ONE submission guidelines, in your Methods section, please provide a more detailed description of your methodology for the fluorescein-staining of eyes used in 'Clinical examination of patients with dry eye'.

"This study was supported by a grant from Otsuka Pharmaceutical Co., Ltd. to Nihon University Itabashi Hospital. Otsuka Pharmaceutical Co., Ltd. had no role in the study design, data collection and analysis, and the decision to publish this manuscript."

We note that you received funding from a commercial source: Otsuka Pharmaceutical Co., Ltd.

"I have read the jpornal's policy and the authors of this manuscript have the following competing interests: J.S. received personal fees from Santen Pharmaceutical Co. Ltd., Senju Pharmaceutical Co. Ltd., Alcon Pharmaeuticals, outside the submitted work. Y.S., N.I., and A.T. declare that they have no conflict of interest."

Reviewers' comments:

Reviewer's Responses to Questions

**Comments to the Author**

1. Is the manuscript technically sound, and do the data support the conclusions?

Reviewer #1: Yes

Reviewer #2: Partly

2. Has the statistical analysis been performed appropriately and rigorously? 

Reviewer #1: I Don't Know

Reviewer #2: Yes

3. Have the authors made all data underlying the findings in their manuscript fully available?

Reviewer #1: Yes

Reviewer #2: Yes

4. Is the manuscript presented in an intelligible fashion and written in standard English?

Reviewer #1: Yes

Reviewer #2: Yes

5. Review Comments to the Author

Reviewer #1: An interesting and useful study into the role of mucins in SS DED and the usefulness of impression cytology in determining changes in these genes.

Some comments/suggestions:

1. Some unusual data obatiend post-therapy at 12 weeks. I agree that the authors should investigate this in future studies by providing a diary or checklist to improve patient compliance. Is it possible that the efficacy of the drugs drops off in extended treatment periods?

2. Please check font size is consistent throughout text, as it varies somewhat.

3. Do the authors have any photographs of cells on schirmer stripos (possible stained) to show how cells look in the 2 groups? Morphological assessments may be useful alongside mRNA data in explaining results observed.

4. No sex-matching was doen between controls and SS pateints. Can the authors briefly explain why they could not use all females to overcome this?

5. What was inclusion/exclusion criteria for controls? Only detail given for SS patients.

6. What were the cut-off points used for TBUT for normals?

7. Can authors explain why they stored mRNA at 4oc rather than at -20/-80 prior to extraction? RNA is notoriously unstable, even at -20.

8. How did authors decide on the most appropriate control gene, GAPDH? Even control genes may be altered post-therapy (i.e. actin etc). Did they consider alternatives?

9. If previous mucin studies have sampled from the bulbal conjunctivae of SS-DE patients, why did the authors choose to sample from the upper palpebral conjunctivae?

10. Related to 9, usually SS patients are considered moderate-to-severe dry eye, due to the extended diagnosis times. Is this an early-diagnosis group to be considered mild-to-moderate? Or are more clinical assessments needed to confirm this diagnosis?

Reviewer #2: This work evaluated the mucin-related gene expression before and after rebamipide treatment in SSDE patients with the assistant of impression cytology. Following is my comments:

1. The tile focuses on the “A method”, but actually the primary and secondary objective of this study focused on the mucin-related gene expression. I think the title should be considered to modify.

2. The technique of impression cytology should be described more detailed including the position on the upper tarsal conjunctiva, the duration touched on it and how to press the paper. And usually the impression cytology is performed on the bulbar conjunctiva, why the authors choose the tarsal conjunctival?

3. Did the SSDE patients use only rebamipide during this study? The authors should state it clearly. This may associate with the unusual change of the data in week 12.

4. The data of this study showed significantly difference of mucin mRNA expression between the male and female controls. So, I think when compares the baseline mucin level, the authors can only use the female data in the control group to compare with SS group, because there is no male in SS group. Current results showed there was no difference of the mucin expression between controls and the baseline of SS group which was not consistent with most studies. When the authors removed the influence of gender, the results may change.

5. It will be better if the authors could provide representative fluorescein staining photos before and after treatment.

6. PLOS authors have the option to publish the peer review history of their article (what does this mean?). If published, this will include your full peer review and any attached files.

Reviewer #1: **Yes: **Suzanne Hagan

Reviewer #2: No

---

## [Author Response · Author response to Decision Letter 0]

30 Oct 2020

Authors’ responses to the reviewers’ comments

We express our sincere appreciation to the reviewers for their insightful comments, which have helped us to significantly improve our manuscript. We have addressed all comments below in a point-by-point manner (reviewers’ comments are shown in bold-face and italics).

Reviewer #1

1. Some unusual data obatiend post-therapy at 12 weeks. I agree that the authors should investigate this in future studies by providing a diary or checklist to improve patient compliance. Is it possible that the efficacy of the drugs drops off in extended treatment periods?

Responses: We agree with your supposition and believe that the reduction in therapeutic efficacy associated with long-term use of rebamipide ophthalmic suspension should be evaluated. For this paper, we have added a picture—Figure 6—of a representative patient in whom the treatment was efficacious. However, individual differences in the efficacy of rebamipide ophthalmic suspension is a question for future research. We have added the aforementioned information on lines 353–354 in the manuscript.

2. Please check font size is consistent throughout text, as it varies somewhat.

Responses: We appreciate your attention to detail. We have determined the font size according to the submission rules of PLOS ONE and standardized it throughout the manuscript.

3. Do the authors have any photographs of cells on schirmer stripos (possible stained) to show how cells look in the 2 groups? Morphological assessments may be useful alongside mRNA data in explaining results observed.

Responses: In impression cytology using filter paper (e.g., Shirmer’s test paper), it is difficult to examine the cells adhering to the filter paper by histological examination because the filter is stained by the staining fluid and does not become transparent. However, we believe that examining the type of cells collected by impression cytology is important to make clinical sense of our test results.

4. No sex-matching was doen between controls and SS pateints. Can the authors briefly explain why they could not use all females to overcome this?

Responses: We agree with your point; accordingly, we have revised the results to compare the SS-DE group with age- and sex-matched controls. We also revised Table 4 and Figure 5. Furthermore, we have revised the descriptions on Page 19, line 249–258.

5. What was inclusion/exclusion criteria for controls? Only detail given for SS patients.

Responses: The inclusion/exclusion criteria for the control group are described on Page 7, lines 90 to 92 as follows: “The control group consisted of 33 healthy adult volunteers (7 men and 26 women) without ocular disorders and who did not wear contact lenses.”

6. What were the cut-off points used for TBUT for normals?

Responses: The statement that the normal range for TBUT is longer than 5 seconds has been added on page 8, lines 117–118.

7. Can authors explain why they stored mRNA at 4oc rather than at -20/-80 prior to extraction? RNA is notoriously unstable, even at -20.

Responses: We placed the filter papers in RNAlater solution immediately after collecting the specimens. The mRNA extraction was performed on the day of sample collection. In preliminary experiments, we observed that mRNA extraction was better without repeated freeze-thaw cycles; therefore, it was stored at 4°C until extraction.

8. How did authors decide on the most appropriate control gene, GAPDH? Even control genes may be altered post-therapy (i.e. actin etc). Did they consider alternatives?

Responses: We agree that this is an important point in performing quantitative PCR by the ΔΔCT method. GAPDH is a commonly used internal control. We compared the mRNA expression of actin with that of GAPDH, but there was no difference between the two. In the future, however, it would be desirable to create a calibration curve with a pre-determined standard mRNA.

9. If previous mucin studies have sampled from the bulbal conjunctivae of SS-DE patients, why did the authors choose to sample from the upper palpebral conjunctivae?

Responses: We performed impression cytology with filter paper as it has a lower specimen collection volume than with nitrocellulose membranes. Therefore, the upper palpebral conjunctiva was selected as the area where sufficient pressure could be applied to the filter paper. Future investigations are needed to examine optimal collection sites.

10. Related to 9, usually SS patients are considered moderate-to-severe dry eye, due to the extended diagnosis times. Is this an early-diagnosis group to be considered mild-to-moderate? Or are more clinical assessments needed to confirm this diagnosis?

Responses: This is an early-diagnosis group. As described in the Discussion section, systemic anti-inflammatory treatment for primary disease may have altered the severity of SS-DE (lines 317–319).

Reviewer #2

1. The tile focuses on the “A method”, but actually the primary and secondary objective of this study focused on the mucin-related gene expression. I think the title should be considered to modify.

Responses: We have changed the title as following: “Assessment of mucin-related gene changes following treatment with rebamipide ophthalmic suspension in Sjögren’s syndrome-associated dry eye.”

2. The technique of impression cytology should be described more detailed including the position on the upper tarsal conjunctiva, the duration touched on it and how to press the paper. And usually the impression cytology is performed on the bulbar conjunctiva, why the authors choose the tarsal conjunctival?

Responses: We have expanded the description about impression cytology as following: “the 5 mm tip of Schirmer’s test paper (Tear Production Measuring Strips, AYUMI Pharmaceutical Corporation, Tokyo, Japan) was placed at the center of the upper palpebral conjunctiva (Fig 1b) for 5 seconds using Beaupre cilia forceps” (pages 10, lines 138–139).

Impression cytology using filter paper is considered to have a lower specimen collection volume than that with nitrocellulose membranes. Therefore, we selected the upper palpebral conjunctiva as the area where sufficient pressure could be applied to the filter paper. Furthermore, we felt that the upper palpebral conjunctiva was suitable for the investigation of mucin-related genes because this region has a high density of goblet cells. However, in the present study, these features may have caused the difference between the control and SS-DE groups to be less pronounced.

3. Did the SS-DE patients use only rebamipide during this study? The authors should state it clearly. This may associate with the unusual change of the data in week 12.

Responses: We agree with your suggestion and, accordingly, have added following description: 

“The rebamipide eye drops were supplemented with sodium hyaluronate in three patients, and with artificial tears and sodium hyaluronate in one; no additional medications were added for nine patients. The compositions of the eye drop were not changed during the observation period.” (page 15, lines 205–208).

4. The data of this study showed significantly difference of mucin mRNA expression between the male and female controls. So, I think when compares the baseline mucin level, the authors can only use the female data in the control group to compare with SS group, because there is no male in SS group. Current results showed there was no difference of the mucin expression between controls and the baseline of SS group which was not consistent with most studies. When the authors removed the influence of gender, the results may change.

Responses: We have revised the results to comparing the SS-DE group with age- and sex-matched controls. We have also revised Table 4 and Figure 5. Furthermore, we have revised the descriptions on Page 19, line 249–258.

5. It will be better if the authors could provide representative fluorescein staining photos before and after treatment.

Responses: We have added Figure 6, which shows the eyes of a representative case.

---

## [Editor Report · Decision Letter 1]

6 Nov 2020

Assessment of mucin-related gene alterations following treatment with rebamipide ophthalmic suspension in Sjögren’s syndrome-associated dry eyes

PONE-D-20-24941R1

Dear Dr. Yamagami,

We’re pleased to inform you that your manuscript has been judged scientifically suitable for publication and will be formally accepted for publication once it meets all outstanding technical requirements.

Kind regards,

Alfred S Lewin, Ph.D.

Section Editor

PLOS ONE
---

## [Editor Report · Acceptance letter]

10 Nov 2020

PONE-D-20-24941R1 

Assessment of mucin-related gene alterations following treatment with rebamipide ophthalmic suspension in Sjögren’s syndrome-associated dry eyes 

Dear Dr. Yamagami:

I'm pleased to inform you that your manuscript has been deemed suitable for publication in PLOS ONE. Congratulations! Your manuscript is now with our production department. 

Kind regards, 

on behalf of

Dr. Alfred S Lewin 

Section Editor

PLOS ONE